# Effect of Mn Substitution Fe on the Formability and Magnetic Properties of Amorphous Fe$_{88}$Zr$_8$B$_4$ Alloy

Xin Wang [1], Qiang Wang [2,3], Benzhen Tang [2,3], Ding Ding [2,3,*], Li Cui [1] and Lei Xia [2,3,*]

1. College of Engineering, Shanghai Polytechnic University, Shanghai 201209, China; wangx@sspu.edu.cn (X.W.); cuili@sspu.edu.cn (L.C.)
2. Institute of Materials, Shanghai University, Shanghai 200072, China; mat_wq@shu.edu.cn (Q.W.); bz.tang@cqnu.edu.cn (B.T.)
3. Center for Advanced Microanalysis, Shanghai University, Shanghai 200444, China
* Correspondence: d.ding@shu.edu.cn (D.D.); xialei@shu.edu.cn (L.X.); Tel.: +86-021-66135067 (D.D. & L.X.)

**Abstract:** Elemental substitution is commonly used to improve the formability of metallic glasses and the properties of amorphous alloys over a wide compositional range. Therefore, it is essential to investigate the influence of element content change on the formability as well as magnetic and other properties. The purpose is to achieve tailorable properties in these alloys with enhanced glass forming ability. In this work, the glass-forming ability (GFA) and magnetic properties of the minor Mn-substituted Fe$_{88}$Zr$_8$B$_4$ amorphous alloy were investigated. The addition of Mn improving the amorphous forming ability of the alloy. With the addition of Mn, the magnetic transition temperature, saturation magnetization and the magnetic entropy changes ($-\Delta S_m$) peaks decreased simultaneously, which is possibly caused by the antiferromagnetic coupling between Fe and Mn atoms. The dependence of $-\Delta S_m{}^{peak}$ on $T_c$ displays a positive correlation compared to the $-\Delta S_m{}^{peak}$- $T_c{}^{-2/3}$ relationship proposed by Belo et al.

**Keywords:** metallic glasses; glass formability; elemental substitution; magnetocaloric effect





## 1. Introduction

With the deterioration of the global energy crisis and environmental pollution, it has become more and more urgent to seek clean energy, renewable energy and energy-saving technologies in recent years [1,2]. The traditional refrigeration system using gas compression/expansion technology is a large energy consumer, low efficiency and not environmentally friendly due to ozone-depleting chlorofluorocarbons. The magnetic refrigerator using a solid refrigerant is more energy efficient (up to 30% reduction of energy loss), more environmentally friendly (free of freon) and has a larger energy density (conducive to miniaturization) when compared to the freon compression machine [2,3]. Therefore, it is essential to find materials with excellent magnetocaloric properties because magnetic refrigeration technology is developed based on the magnetocaloric effect (MCE) of these magnetic refrigerants [4].

The MCE characteristics of the prepared materials so far are usually evaluated by two factors: maximum magnetic entropy change ($-\Delta S_m{}^{peak}$) and refrigeration capacity (*RC*). Intermetallic compounds, such as Gd$_5$(Si$_x$Ge$_{1-x}$)$_4$, La(Fe,M)$_{13}$ (M = Si,Co,Al), MnFe(P$_{1-x}$As$_x$), NiMnGa and LaMnO$_3$ [5–10], exhibit a giant $-\Delta S_m{}^{peak}$ but relatively low *RC* due to their sharp and narrow magnetic entropy change ($-\Delta S_m$) peaks. Additionally, the MCE of the intermetallic compounds is closely related to phase structure, which means the Curie temperature ($T_c$) of these intermetallic refrigerants can hardly be tuned without deteriorating their $-\Delta S_m{}^{peak}$, just like the tunable mechanical properties in the Zr-Cu system [11]. On the other hand, metallic glasses, such as Gd-based and Fe-based amorphous alloys [12–25], usually exhibit huge *RC* but a relatively low $-\Delta S_m{}^{peak}$ due to their broad $-\Delta S_m$ humps. In addition, the $T_c$ and $-\Delta S_m{}^{peak}$ of these amorphous alloys are tailorable

by randomly selecting the composition of the alloys within a certain composition range, such as the compositional-induced structural change in the ZrNi system [26].

It has been found that the constant $-\Delta S_m$ peak in a long temperature range above and below the Curie temperature, namely, a table-like $-\Delta S_m$ profile, is the most ideal form for magnetic refrigerants working in an Ericsson cycle [16–20]. Obviously, an approximately flat $-\Delta S_m$ profile is difficult to achieve in intermetallic compounds not only because of their narrow $-\Delta S_m$ peak but also due to the difficulty of tuning their $T_c$ without dramatically decreasing their $-\Delta S_m{}^{peak}$. In contrast, a table-like $-\Delta S_m$ profile can be easily achieved in amorphous composites with an appropriate Curie temperature and $-\Delta S_m$ peak due to their tunable $T_c$ and $-\Delta S_m{}^{peak}$ over a wide compositional range.

Microalloying and elemental substitution are the usual methods used to improve the formability of metallic glasses and to change the properties of amorphous alloys over a wide compositional range [14,15,19–21], and therefore, they are very important for achieving tailorable properties in these based alloys, accompanied by better glass-forming ability. In this study, we prepared $Fe_{88}Zr_8B_4Mn_x$ as-spun ribbons by adding Mn to lower the Fe content in based $Fe_{88}Zr_8B_4$ samples. The glass formability and magnetic properties of the $Fe_{88}Zr_8B_4Mn_x$ metallic glasses were studied in comparison with those of the $Fe_{88}Zr_8B_4$ glassy alloy. Based on these results, the effect of Mn alloying on the formability and MCE of the original base alloys, accompanied by the interaction mechanism between the internal elements, were investigated.

## 2. Materials and Methods

A series of button-shaped metal samples with element compositions of $Fe_{88-x}Zr_8B_4Mn_x$ (x = 2, 5, 8, 10) were prepared by heating the high-purity raw materials until fully integrated at least four times in a non-consumable electrode high vacuum arc melting furnace that filled with high purity argon after the oxygen absorption protection operation of the titanium ingot. The $Fe_{88-x}Zr_8B_4Mn_x$ ribbons (typically 2–3 mm wide and ~40 μm thick) were prepared in a protective atmosphere of argon by ejecting the melts from the quartz tube on a rotating copper wheel under a pure Ar atmosphere. The structural characteristics of the obtained alloys were ascertained by means of X-ray diffraction (XRD) on a Rigaku diffractometer (model D/max-2550, Tokyo, Japan) using $K_\alpha$ radiation of copper, and the thermodynamic parameters were tested on the differential scanning calorimetry (DSC) on a NETZSCH calorimeter (model 404C, Selb, Germany) with the heating curves obtained at 0.333 K/s. Magnetic measurements were performed on the vibrating sample magnetometer (VSM) of a Physical Properties Measurement System (PPMS model 6000, Quantum Design, San Diego, CA, USA), and the magnetization vs. temperature and the isothermal magnetization curves were obtained.

## 3. Results and Discussion

### 3.1. Descriptive of the Experimental Results

The preliminary structure verification information of the prepared $Fe_{88-x}Zr_8B_4Mn_x$ (x = 2, 5, 8, 10) ribbons characterized by the broadened diffraction hump at about 43°, as shown in the curves in the XRD patterns in Figure 1, roughly explains the amorphous trait of the samples. From another point of view, the glassy feature, shown in Figure 2a, was further ascertained on the DSC curves that displayed a clear endothermic glass transition point that appeared before the crystallization peak. Figure 2b illustrates the melting behaviors of the $Fe_{88-x}Zr_8B_4Mn_x$ (x = 2, 5, 8, 10) alloys. The glass transition temperature ($T_g$), crystallization temperature ($T_x$) and liquid temperature ($T_l$) of the glassy ribbons obtained from their DSC traces are listed in Table 1. For comparison, in Table 1 the thermodynamic properties of the $Fe_{88}Zr_8B_4$ glassy ribbon have been recalculated. As such, the reduced glass transition temperature ($T_{rg} = T_g/T_l$) and the parameter $\gamma$ ($= T_x/(T_g + T_l)$), both of which are usually applied as the major reference for the glass formability (GFA) of the alloys [27,28], can be obtained accordingly to be 0.503 and 0.353 for $Fe_{86}Zr_8B_4Mn_2$, 0.506 and 0.355 for $Fe_{83}Zr_8B_4Mn_5$, 0.517 and 0.353 for $Fe_{80}Zr_8B_4Mn_8$ and 0.512 and 0.356

for $Fe_{78}Zr_8B_4Mn_{10}$. The GFA of the $Fe_{88}Zr_8B_4$ glassy alloy was generally enhanced by Mn substitution for Fe, which agrees well with the multicomponent rule for glass forming [4]. The glass formation enthalpy ($\Delta H^{amor}$) of the $Fe_{88-x}Zr_8B_4Mn_x$ (x = 0, 2, 5, 8, 10) alloys were calculated using Miedema's model, which can be used as a reference to investigate the mechanism of alloy's GFA in more detail by comparing the internal bonding changes after the addition of new elements [28]. The calculated $\Delta H^{amor}$ shows a decreasing trend with a value of about −2.85 kJ/mol for x = 0, −3.26 kJ/mol for x = 2, −3.84 kJ/mol for x = 5, −4.68 kJ/mol for x = 8 and −5.13 kJ/mol for x = 10. As the formability of metallic glass is closely related to $\Delta H^{amor}$, the enhanced GFA by Mn addition is most likely attributed to a decrease in $\Delta H^{amor}$ as the Mn content increases. The composition of alloys was ascertained by chemical analysis. In order to ascertain the homogeneity of the samples, we selected an amorphous ribbon with lower glass formability, that is, the $Fe_{78}Zr_8B_4Mn_{10}$ ribbon, for HRTEM examination. Figure 3 shows the HRTEM image of the $Fe_{78}Zr_8B_4Mn_{10}$ ribbon. The ribbon is homogeneous with a disordered atomic configuration.

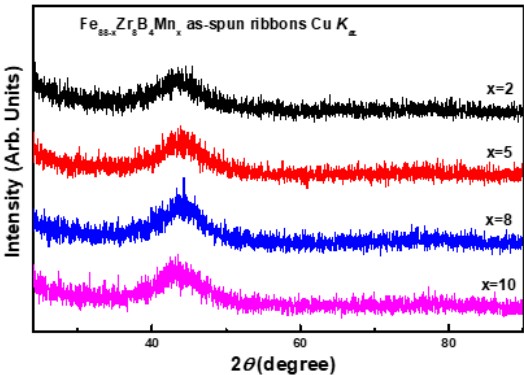

**Figure 1.** XRD patterns of the $Fe_{88-x}Zr_8B_4Mn_x$ (x = 2, 5, 8, 10) as-spun ribbons.

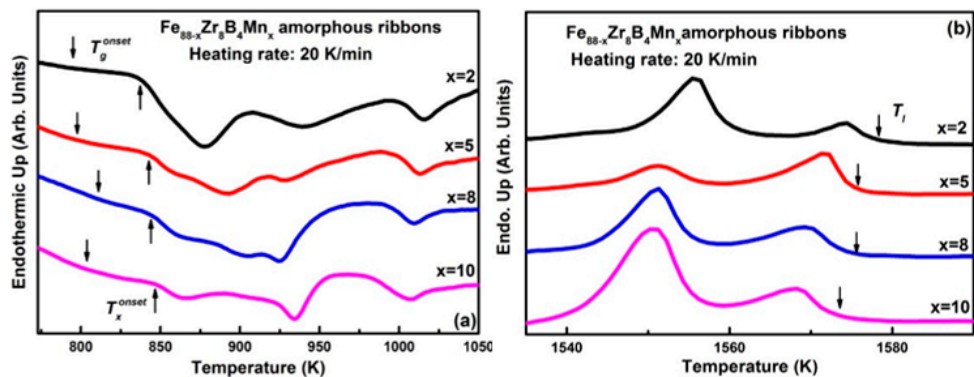

**Figure 2.** DSC traces of the $Fe_{88-x}Zr_8B_4Mn_x$ (x = 2, 5, 8, 10) alloys with the heating curves obtained at 0.333 K/s. (**a**) shows the glass transition temperature ($T_g$), crystallization temperature ($T_x$) and (**b**) is the liquid temperature ($T_l$).

**Table 1.** Thermal properties of $Fe_{88-x}Zr_8B_4Mn_x$ amorphous samples.

| $Fe_{88-x}Zr_8B_4Mn_x$ | $T_g$ (K) | $T_x$ (K) | $T_l$ (K) | $T_{rg}$ | $\gamma$ | Ref. |
|---|---|---|---|---|---|---|
| x = 0 | 787 | 840 | 1611 | 0.489 | 0.350 | [19] |
| x = 2 | 794 | 837 | 1578 | 0.503 | 0.353 | |
| x = 5 | 797 | 842 | 1575 | 0.506 | 0.355 | Present work |
| x = 8 | 811 | 843 | 1575 | 0.515 | 0.353 | |
| x = 10 | 806 | 847 | 1573 | 0.512 | 0.356 | |

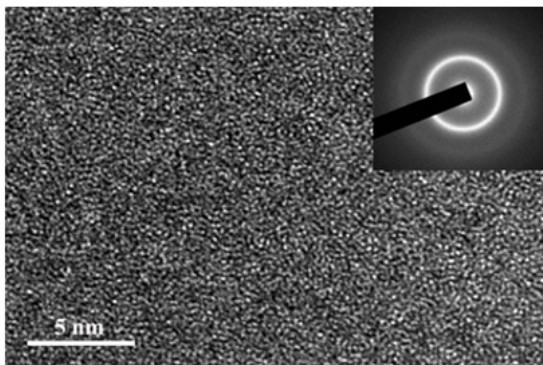

**Figure 3.** The HRTEM image of the $Fe_{78}Zr_8B_4Mn_{10}$ amorphous alloy.

The relationship between magnetization (*M*) and temperature (*T*) of the amorphous $Fe_{88-x}Zr_8B_4Mn_x$ samples was measured, respectively, from 200 to 380K after the operation by a zero-field cooling method. By performing a derivative process on the curves in Figure 4, the Curie temperature of a series of $Fe_{88}Zr_8B_4$ samples with Mn addition was found to be 283 K for x = 2, 263 K for x = 5, 238 K for x = 8 and 225 K for x = 10. The $T_c$ of the metallic glasses with Mn added, as seen in the embedded image in the upper right corner of Figure 4, had a decreasing trend compared with the $Fe_{88}Zr_8B_4$ samples. It is known that the magnetic properties of Fe-based amorphous alloys exhibit a close correlation with the direct 3d interactions of Fe atoms [19–23]. As with the situation shown in the FeZrB series amorphous alloys, although the reduction of Fe content reduces the number of interactions between 3d atoms, the introduction of other atoms can enhance the 3d interaction between Fe to some extent and thereby enhance the magnetic properties of the FeZrB amorphous alloys. A completely different trend of the Mn addition on the Fe-Zr-B amorphous alloys, however, induced antiferromagnetic coupling between Fe and Mn atoms [29], which deteriorated the Fe-Fe interactions and therefore led to a decrease in the Curie temperature of the $Fe_{88}Zr_8B_4$ metallic glass.

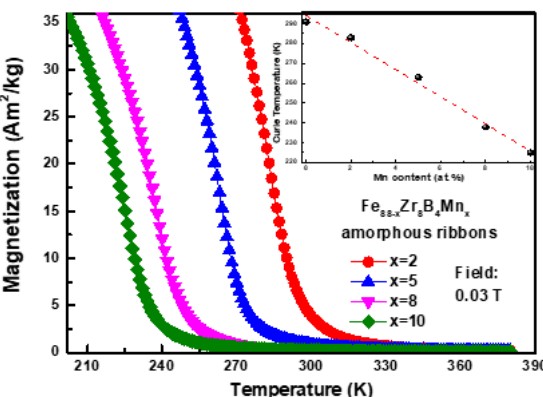

**Figure 4.** The temperature dependence of the magnetization (M-T) curves of the $Fe_{88-x}Zr_8B_4Mn_x$ samples, inset the compositional dependence of $T_c$ in $Fe_{88-x}Zr_8B_4TM_x$ (TM=Co, Mn) metallic glasses.

Figure 5 shows the hysteresis loops of the amorphous $Fe_{88-x}Zr_8B_4Mn_x$ (x = 0, 2, 5, 8 and 10) samples measured at 200 K under 5 Tesla. The ribbons are all soft magnetic and exhibit negligible coercivity, indicating that the new series of FeZrB alloys is similar to the based alloys that are potential materials for practical application because they can easily be magnetized and demagnetized. The saturation magnetization ($M_s$) of the $Fe_{88-x}Zr_8B_4Mn_x$ glassy ribbons is about 109 $Am^2/kg$ for x = 0, 94.6 $Am^2/kg$ for x = 2, 80.3 $Am^2/kg$ for x = 5, 65.4 $Am^2/kg$ for x = 8 and 58.33 $Am^2/kg$ for x = 10. To facilitate observation, Figure 5 plots the variation curve of $M_s$ and Mn content inset in the lower right corner. The $M_s$ of

the samples decreased dramatically as the Mn content increased, which indicates reduced magnetic moments of the glassy samples induced by antiferromagnetic coupling between Fe and Mn atoms.

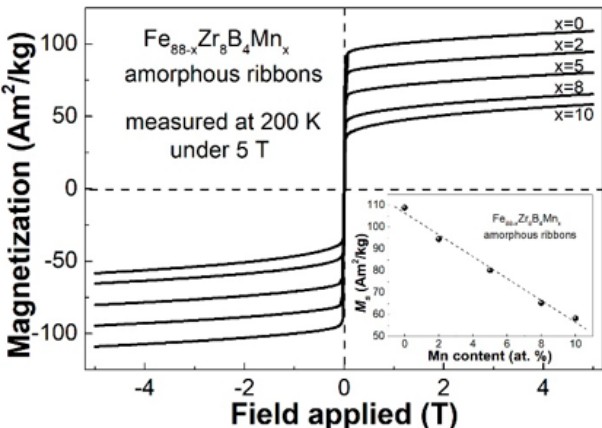

**Figure 5.** The hysteresis loops of the amorphous $Fe_{88-x}Zr_8B_4Mn_x$ (x = 0, 2, 5, 8 and 10) samples measured at 200 K under 5 Tesla. The inset shows the relationship between the saturation magnetization ($M_s$) of the $Fe_{88-x}Zr_8B_4Mn_x$ amorphous ribbons and the Mn content.

Considering that the saturation magnetization is closely related to the magnetic moment of the material, the reduced magnetic moments by Mn addition may also result in the deteriorated magnetocaloric properties of the glassy $Fe_{88-x}Zr_8B_4Mn_x$ samples. By measuring the isothermal *M-H* curves (with *H* = 5 T in the present work) of different samples, a series of different Arrott plots (converted from the *M-H* curves) under different temperatures was obtained to investigate the magnetic phase change character. The Arrott plots of the $Fe_{88-x}Zr_8B_4Mn_x$ (x = 2, 5, 8 and 10) glassy samples, as shown in Figure 6, display a positive slope with a nearly C-shape, which demonstrates a second-order magnetic phase transition (MPT) during the ferromagnetic–paramagnetic transition in all the samples [30].

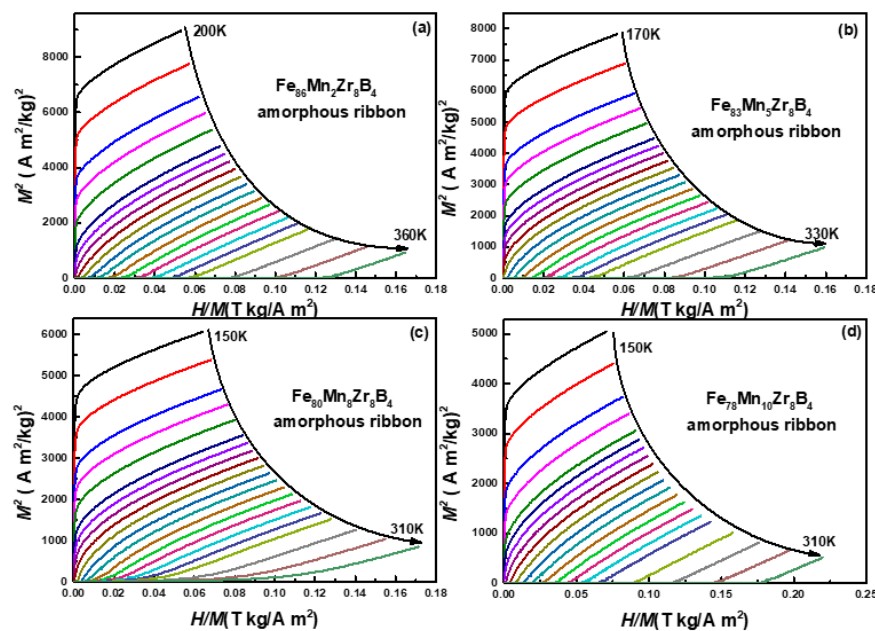

**Figure 6.** The Arrott plots of the $Fe_{88-x}Zr_8B_4Mn_x$ (x = 2, 5, 8 and 10) glassy samples. (**a**) for $Fe_{86}Zr_8B_4Mn_2$, (**b**) for $Fe_{83}Zr_8B_4Mn_5$, (**c**) for $Fe_{80}Zr_8B_4Mn_8$ and (**d**) for $Fe_{78}Zr_8B_4Mn_{10}$.

According to Maxwell's relations, we can calculate the magnetic entropy change $(-\Delta S_m)$ of these amorphous alloys.

$$\Delta S_m(T, H) = S_m(T, H) - S_m(T, 0) = \int_0^H \left(\frac{\partial M}{\partial H}\right)_H dH \qquad (1)$$

The $-\Delta S_m$-T curves of the amorphous $Fe_{88-x}Zr_8B_4Mn_x$ samples are plotted in Figure 7: (a) for $Fe_{86}Zr_8B_4Mn_2$, (b) for $Fe_{83}Zr_8B_4Mn_5$, (c) for $Fe_{80}Zr_8B_4Mn_8$ and (d) for $Fe_{78}Zr_8B_4Mn_{10}$ to further study the MCE of the alloys. The samples exhibited a typical lambda shape with a maximum in the vicinity of $T_c$. The wide $-\Delta S_m$ peak of the series of $Fe_{88-x}Zr_8B_4Mn_x$ glassy ribbons is in accordance with the characteristics of the second-order MPT [12–14,19–21]. The $-\Delta S_m^{peak}$ of the $Fe_{88-x}Zr_8B_4Mn_x$ glassy ribbons that increased from 1 to 5T is displayed in Figure 7. As shown in Table 2, the MCE behavior of the $Fe_{88-x}Zr_8B_4Mn_x$ samples can be described by the $-\Delta S_m \propto H^n$ relationship [24,25]. Fitting the relationship curve of the $-\Delta S_m$ and $H$ after logarithmic processing, the $n$ exponent of the $Fe_{88-x}Zr_8B_4Mn_x$ alloys under different temperatures was ascribed from the fitted curves, as the value plotted in the $n$-T curves in Figure 7e. All the amorphous samples displayed the typical magnetocaloric behaviors of soft magnetic metallic glasses: $n$ is nearly 1 at low temperatures when the sample is ferromagnetic, then gradually reduced to a minimum value near $T_c$ and finally increased dramatically to a value up to 2 at the paramagnetic range. The $n$ in the vicinity of $T_c$, as listed in Table 2, is about 0.769 for $Fe_{88}Zr_8B_4$, 0.745 for $Fe_{86}Zr_8B_4Mn_2$, 0.736 for $Fe_{83}Zr_8B_4Mn_5$, 0.740 for $Fe_{80}Zr_8B_4Mn_8$ and 0.736 for $Fe_{78}Zr_8B_4Mn_{10}$, all of which are roughly in accordance with the ones of other fully amorphous alloys [13,14,20,21,24,25,31,32].

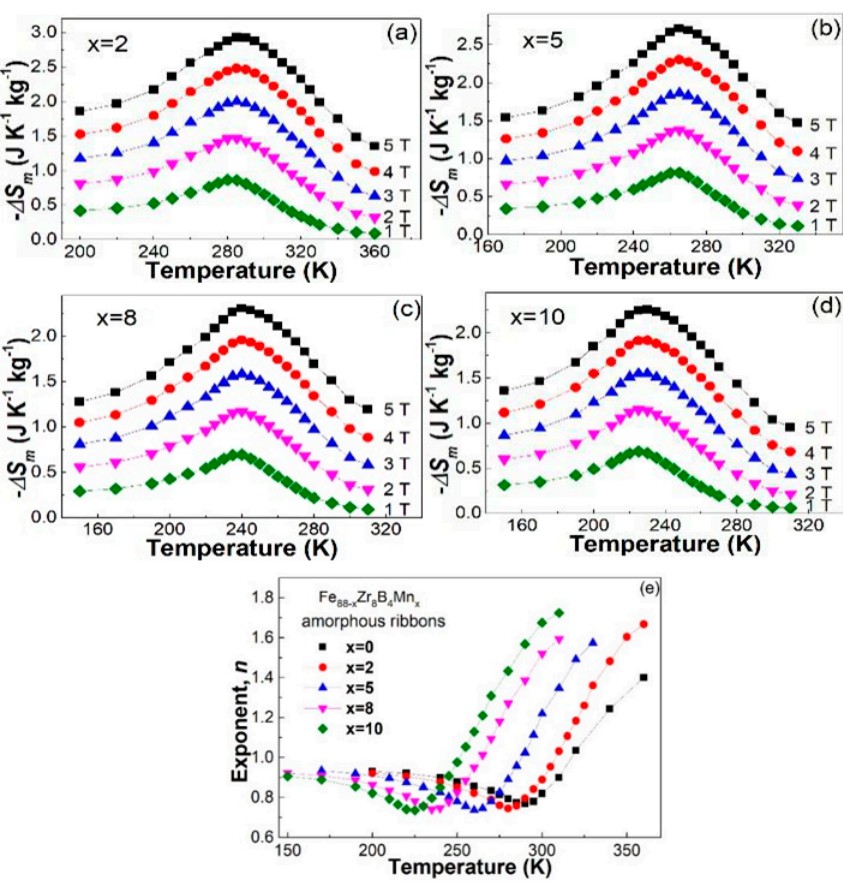

**Figure 7.** The $-\Delta S_m$-T curves of the amorphous $Fe_{88-x}Zr_8B_4Mn_x$ samples from 1 to 5T: (**a**) for $Fe_{86}Zr_8B_4Mn_2$, (**b**) for $Fe_{83}Zr_8B_4Mn_5$, (**c**) for $Fe_{80}Zr_8B_4Mn_8$ and (**d**) for $Fe_{78}Zr_8B_4Mn_{10}$ (**e**) the $n$-T curves of the $Fe_{88-x}Zr_8B_4Mn_x$ samples obtained by the $-\Delta S_m \propto H^n$ relationship.

**Table 2.** Magnetic and magnetocaloric properties of $Fe_{88-x}Zr_8B_4Mn_x$ amorphous samples.

| $Fe_{88-x}Zr_8B_4Mn_x$ | $T_c$ (K) | $-\Delta S_m^{peak}$ (J·kg$^{-1}$·K$^{-1}$) | | | | | $n$ | Ref. |
|---|---|---|---|---|---|---|---|---|
| | | 1 T | 2 T | 3 T | 4 T | 5 T | | |
| x = 0 | 291 | 0.87 | 1.5 | 2.06 | 2.57 | 3.04 | 0.769 | [19] |
| x = 2 | 283 | 0.867 | 1.471 | 1.999 | 2.480 | 2.931 | 0.745 | |
| x = 5 | 263 | 0.818 | 1.380 | 1.865 | 2.306 | 2.715 | 0.736 | Present work |
| x = 8 | 238 | 0.694 | 1.172 | 1.585 | 1.960 | 2.309 | 0.740 | |
| x = 10 | 225 | 0.689 | 1.152 | 1.551 | 1.919 | 2.261 | 0.736 | |

As predicted above, adding the antiferromagnetic element Mn created a new antiferromagnetic coupling and reduced the overall magnetic moments, and thus, deterioration the $-\Delta S_m^{peak}$ of the $Fe_{88-x}Zr_8B_4Mn_x$ metallic glasses is mostly due to the reduced magnetic moments caused by adding the antiferromagnetic element Mn. Figure 8 shows the $-\Delta S_m^{peak}$ *vs* $T_c^{-2/3}$ plots at each magnetic field for the $Fe_{88-x}Zr_8B_4Mn_x$ amorphous samples. Combining the results obtained in other Fe-based metallic glasses [3,18–25], the dependence of $-\Delta S_m^{peak}$ on $T_c$ displays a positive correlation in the Fe-Zr-B-based metallic glasses, which is contrary to the $-\Delta S_m^{peak}$-$T_c^{-2/3}$ relationship proposed by Belo et al. [12–16,33]. This is probably because some magnetic parameters including $T_c$ and magnetization in Fe-based metallic glasses are dominated by the direct coupling between the Fe-based elements. This factor, which enhances the interaction between Fe atoms, simultaneously improves the Curie temperature, $M_s$ and $-\Delta S_m^{peak}$, and vice versa. In contrast, the factors in RE-based or containing amorphous alloys are more complicated in view of the complex interaction of 4*f* layer electrons of RE with other TM electrons. Overall, the combination of the direct interaction between TM atoms and the indirect interactions between RE-RE and RE-TM atoms results in the complex magnetic performance of the RE-TM-based metallic glasses.

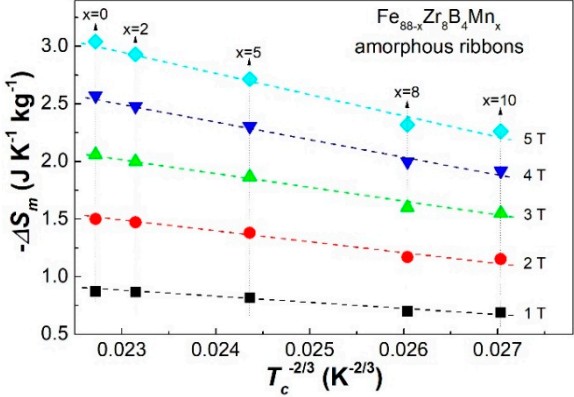

**Figure 8.** The $-\Delta S_m^{peak}$ vs. $T_c^{-2/3}$ plots under a field of 1T, 2T, 3T, 4T, 5T for the $Fe_{88-x}Zr_8B_4Mn_x$ amorphous samples.

### 3.2. Discussion

As predicted above, the $Fe_{88-x}Zr_8B_4Mn_x$ ribbon is homogeneous with a disordered atomic configuration. Adding the antiferromagnetic element Mn creates a new antiferromagnetic coupling and reduces the overall magnetic moments, and thus, the deterioration of the $-\Delta S_m^{peak}$ of the $Fe_{88-x}Zr_8B_4Mn_x$ metallic glasses is mostly due to the reduced magnetic moments caused by adding the antiferromagnetic element Mn. Figure 7 shows the $-\Delta S_m^{peak}$ *vs* $T_c^{-2/3}$ plots at each magnetic field for the $Fe_{88-x}Zr_8B_4Mn_x$ amorphous samples. Combining the results obtained in other Fe-based metallic glasses [3,18–25], the dependence of $-\Delta S_m^{peak}$ on $T_c$ displays a positive correlation in the Fe-Zr-B-based metallic glasses, which is contrary to the $-\Delta S_m^{peak}$-$T_c^{-2/3}$ relationship proposed by Belo et al. [12–16,33]. This is probably because some magnetic parameters including $T_c$ and magnetization in Fe-based metallic glasses are

dominated by the direct coupling between the Fe-based elements. This factor, which enhances the interaction between Fe atoms, simultaneously improves the Curie temperature, $M_s$ and $-\Delta S_m^{peak}$, and vice versa. In contrast, the factors in RE-based or containing amorphous alloys are more complicated in view of the complex interaction of 4*f* layer electrons of RE with other TM electrons. Overall, the combination of the direct interaction between TM atoms and the indirect interactions between RE-RE and RE-TM atoms results in the complex magnetic performance of the RE-TM-based metallic glasses.

### 3.3. Discussion

As predicted above, the adding of antiferromagnetic element Mn creates a new antiferromagnetic coupling and reduces the overall magnetic moments, and thus the deteriorates of the $-\Delta S_m^{peak}$ of the $Fe_{88-x}Zr_8B_4Mn_x$ metallic glasses is mostly due to the reduced magnetic moments by the adding of antiferromagnetic element Mn. Figure 7 shows the $-\Delta S_m^{peak}$ vs. $T_c^{-2/3}$ plots at each magnetic field for the $Fe_{88-x}Zr_8B_4Mn_x$ amorphous samples. Combining the results obtained in other Fe-bases metallic glasses [3,18–25], the dependence of $-\Delta S_m^{peak}$ on $T_c$ displays a positive correlation in the Fe-Zr-B-based metallic glasses which is contrary to the $-\Delta S_m^{peak}$-$T_c^{-2/3}$ relationship proposed by Belo et al. [12–16,33]. This is probably because that some magnetic parameters including $T_c$ and magnetization in Fe-based metallic glasses are dominated by the direct coupling between the based elements Fe. The factor, which enhances the interaction between Fe atoms, will simultaneously improve the Curie temperature, $M_s$ and $-\Delta S_m^{peak}$, and vice versa. In contrast, the factors in RE based or containing amorphous alloys are more complicated in view of the complex interaction of 4*f* layer electrons of RE with other TM electrons. all above, the combination of the direct interaction between TM atoms, indirect interactions between RE-RE and RE-TM atoms makes the complex magnetic performance of the RE-TM based metallic glasses.

### 4. Conclusions

In summary, we obtained the amorphous $Fe_{88-x}Zr_8B_4Mn_x$ (x = 2, 5, 8,10) ribbons using the single roller melt spinning method with an average thickness of about 30~40 μm. The metal glass-forming performance of the $Fe_{88}Zr_8B_4$ alloy was enhanced by substituting Mn for Fe, which is supposed to be related to the alloys, and it was enhanced by the decrease in $\Delta H^{amor}$ with increasing Mn content. To investigate the influence mechanism of Mn substitution on the MCE of the $Fe_{88}Zr_8B_4$ amorphous alloy, the magnetocaloric properties of the $Fe_{88-x}Zr_8B_4Mn_x$ samples and $Fe_{88}Zr_8B_4$ amorphous alloy were systematically analyzed. The Curie temperature, $M_s$ and $-\Delta S_m^{peak}$ were found to decrease simultaneously with the addition of Mn. It is supposed that substituting Mn for Fe, which induced the antiferromagnetic coupling between Fe and Mn atoms, deteriorated the Fe-Fe interactions and therefore lead to a decrease in the $T_c$, $M_s$ and $-\Delta S_m^{peak}$ of the $Fe_{88}Zr_8B_4$ amorphous alloys. The exponent $n$ ($-\Delta S_m \propto H^n$) in the vicinity of $T_c$, or the other magnetic state of the $Fe_{88-x}Zr_8B_4Mn_x$ samples, is similar to the other fully amorphous alloys, which indicates the second-order magnetic phase transition and the amorphous structure of the series amorphous alloys. The dependence of $-\Delta S_m^{peak}$ on $T_c$ displayed a positive correlation in the Fe-based amorphous alloys due to the direct coupling between the Fe-based elements, which is different from the relationship proposed by Belo et al. in rare-earth-containing alloys, in view of the complex interaction of 4f layer electrons of RE with other TM electrons.

**Author Contributions:** Conceptualization, L.X.; methodology, Q.W.; investigation, X.W.; writing—original draft preparation, B.T.; writing—review and editing, D.D.; measurement, L.C.; funding acquisition, L.X. All authors have read and agreed to the published version of the manuscript.

**Funding:** This research was funded by the National Nature Science Foundation of China (Grant Nos. 51271103 and 51671119), the Shanghai Polytechnic University 2021 Youth Academic Backbone Training Project (EGD21QD17) and the Shanghai Educational Science Research Project (C2021062).

**Institutional Review Board Statement:** Not applicable.

**Informed Consent Statement:** Not applicable.

**Data Availability Statement:** Not applicable.

**Acknowledgments:** This research was provided technical support by the Center for Advanced Microanalysis.

**Conflicts of Interest:** The authors declare no conflict of interest.

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
