# Peer review of "Effect of Mn Substitution Fe on the Formability and Magnetic Properties of Amorphous Fe88Zr8B4 Alloy"

_metals, doi:10.3390/met11101577_

Round 1

Reviewer 1 Report

In this manuscript, the authors remark the influence of Mn substitution for Fe on the formability and magnetic properties of an amorphous Fe-Zr-B alloy.

Comments:

  1. The final composition should be checked. Only nominal composition was given. During the process a shift of the composition is a possibility. Likewise, the composition homogeneity of the samples should be also checked with a microanalysis technique.

  1. XRD: The ratio noise/signal should be reduced. From the XRD diffraction patterns provided in Figure 1 is impossible to assure the no existence of a minor amount of a crystalline phase. If it was possible, I also recommend XRD after annealing to check the phase forms at least in the first crystallization process detected by calorimetry (DSC).

  1. The authors should remark the interest of Mn substitution.

  1. The average thickness values 30-40 micrometers variation was done in the same sample or in different samples? It was found an influence of composition?

  1. The authors should improve discussion. I recommend reading articles from 2019 to 2021 because recent references are those of the authors of the manuscript.

Reviewer 2 Report

The authors provide a paper dealing with the influence of Mn substitution on the formability and magnetic properties of Fe88Zr8B4 alloy. The paper can be of interest for Metals. However, MAJOR revisions are requested:

  1. The authors must improve the English. The title is long and difficult to follow and the same is for the abstract. I suggest a new reading to make both clearer and effective.
  2. Amorphous alloys are also known for their good mechanical properties especially in thin film form or ribbon as reported in doi.org/10.1016/j.actamat.2021.116955. The authors must comment on this paper and explain that the importance of amorphous alloys rely in a combination of several properties mechanical and magnetic. This make them good candidates for electronics applications.
  3. It would be good if the authors can add selected SEM images and maybe a table with the EDX compositions as a function of the Mn percentage.
  4. I think it can also be useful to have a comparison of magnetic properties without Mn to be add in the series of samples as a reference.
  5. The authors must also discuss in a better why the amorphization phenomena and GFA considering the effect of the cooling rate and composition. A discussion for ZrNi system is reported in doi.org/10.1016/j.jallcom.2013.12.054 the authors should include this paper in their analysis and comment on the effect of Mn addition.

Reviewer 3 Report

This MS is well presented.  The effect of Mn substitution to magnetic properties is interesting.

I found small amounts of English mistakes. Please check again.

Round 2

Reviewer 1 Report

The revised verion of the manuscript take into account the comments of the reviewer.

Reviewer 2 Report

-